



**Comparing water uptake patterns of two plantations using stable isotopes in Chinese Loess**
**Plateau**
Yongsheng Cui[a], Chengzhong Pan[a][*], Lan Ma[b,c], Zhanwei Sun[b]
*a, College of Water Sciences, Beijing Normal University, Beijing 100875, PR China*
*b, Key laboratory of State Forestry Administration on Soil and Water Conservation, School of Soil and Water*
*Conservation, Beijing Forestry University, Beijing 100083, PR China*
*c, Jixian National Forest Ecosystem Observation and Research Station, CNERN, School of Soil and Water*
*Conservation, Beijing Forestry University, Beijing 100083, PR China*
Number of text pages: 27
Number of tables: 2
Number of figures: 6
**Running title: water uptake modes of two plantations**
[*] Correspondence to:
Dr. Chengzhong Pan
College of Water Sciences, Beijing Normal University
Xinjiekouwai Street 19, Beijing 100875, P.R. China;
Tel: ++86-10-58802736
Fax: ++86-10-58802739
E-mail: pancz@bnu.edu.cn



Abstract

Understanding the water consumption mechanism of plantations is of great significance for the selection of afforestation trees and ecologically sustainable watershed management in semi-arid areas. In this study, *Robinia pseudoacacia* and *Pinus tabulaeformis* plantations, which have been widely planted in Chinese Loess Plateau, were selected to investigate the possible difference in water uptake modes. The spatial and temporal variations in precipitation, xylem and soil water stable isotope compositions ($\delta^2$H, $\delta^{18}$O) in 2019–2020 were analyzed, and the water uptake modes of plantations were quantified using the direct inference approach and MixSIAR model, with contrasting soil moisture dynamics. The results showed that $\delta^{18}$O values were positively correlated with air temperature and negatively correlated with precipitation volume. The $\delta^{18}$O content of surface soil (0–40 cm) closely related precipitation input, while those of deep soil layers (100–200 cm) remained stable. When compared with the direct inference approach, the MixSIAR model performed more effectively in quantifying water apportionment, especially in a drought year. In a drought year, *R. pseudoacacia* showed strong drought resilience to absorb water from deep soil, the soil layers of 0–40 cm and 40–200 cm contributed 32.9% and 67.1% to water absorption of *R. pseudoacacia*, and 62.2% and 37.8% to that of *P. tabulaeformis*, respectively, while there appeared to be minor differences in soil water uptake between the two plantations in a humid year. Generally, *R. pseudoacacia* consumed more water than *P. tabulaeformis*, especially in a humid year, and the former inclined to absorb soil layer with enriched soil moisture. The results indicated that *R. pseudoacacia* plantation may increase transpiration and cause dried deep soil layers when compared with *P. tabulaeformis*. This study improves our understanding of water uptake mechanisms of plantations and helps with selection of suitable plant species for ecological management in Chinese Loess Plateau.

Keywords

Root water uptake; Stable isotope; Soil water storage; Plantation; MixSIAR model

1. Introduction



Soil moisture is the main water source of vegetation growth, especially for the Loess Plateau in China with
deep soil layers. It has been proved that large-scale afforestation programs since the 1990s, especially plantations,
could increase transpiration and soil water consumption, leading to increased potential for soil desiccation (Liu et
al., 2018; Jia et al., 2017; Ma et al., 2014). Therefore, it is urgent and necessary to identify a reasonable vegetation
restoration scheme to protect the fragile ecosystems in the Loess Plateau.
Plants absorb soil water through root systems and transport it to stem and leaves, where it then dissipates into
the atmosphere. Traditionally, the variation in soil water content (SWC) in different soil layers was measured to
estimate plant water utilization, and the dynamics of soil water storage helps us to understand the relationship
between root water uptake (RWU) and soil moisture (Qiu et al., 2021; Fang et al., 2016; Jackisch et al., 2020). The
soil layers of 0–200 cm was classified as an active layers based on the temporal change in SWC (Wang et al., 2015).
However, this method cannot accurately depict all water sources and is restricted by the labor-intensive nature of
these measurements.
The stable isotopic technique shed light on the study of plant water uptake patterns. As the "fingerprint of water"
(Meißner et al., 2013), the natural stable isotopes of hydrogen and oxygen in water serve as important recorders of
hydrological and ecological processes, providing solid information to explore the water transformation between
different water pools, e.g., atmosphere, soil and plant (West et al., 2006; Dai et al., 2020; Barnes and Allison, 1988;
Mahindawansha et al., 2018). Furthermore, Geris et al. (2017) proved that for most species, there is no isotopic
fractionation during the process of RWU and transportation along the conduit before transpiration, making it
possible to specify plant water sources and water uptake patterns.
The water transport process through the soil-plant atmosphere continuum (SPAC) has been widely studied in
the context of water phase transition and movement and in many contexts. Compared to *Artemisia gmelinii*, *Vitex*
*negundo* obtained more water from deeper soils as the water stress increased, showed more flexibility to acclimate
the drought (Wang et al., 2017). Precipitation, soil water and stem water were sampled for analyses, Chang et al.



(2019) studied the water use strategies of four vegetation types after succession in the Loess Plateau and found that
the soil water used by different vegetation types both extended deeper with the development of succession, and
water was accessed from deeper and shallower soil during dry and wet seasons, respectively. Wang et al. (2021a)
investigated water use characteristics of *Robinia pseudoacacia* in plantations of 18 and 30 years, and found 30-yr
*R. pseudoacacia* mainly took water from shallow soil, while 18-yr *R. pseudoacacia* displayed variable responses to
variations in precipitation. These studies mainly focused on plants with yearly leaf abscission and studied the
seasonal variations generally, while studies concentrated in trees are relatively rare, especially comparisons between
different species, and there is a lack of understanding of the responses under different hydrological conditions. Thus,
investigation of the mainly artificially planted trees, such as *R. pseudoacacia* and *Pinus tabulaeformis*, in the Loess
Plateau would provide a scientific basis for forest management.

Three classes of approaches have been developed and used to determine the proportion of water sources

accessed by plants, which is the direct inference approach (Amin et al., 2020), the linear mixing models (Evaristo
et al., 2017) and Bayesian mixing models. Among them, the MixSIAR model (Bayesian mixing model) exhibited
good performance and prevailed in the assessment of plant water source apportionment (Beyer et al., 2018; Duvert
et al., 2021). The MixSIAR model has been directly applied to the Loess Plateau directly in many studies (Wang et
al., 2019; Tao et al., 2021; Wu et al., 2021), but has rarely been contrasted with other methods, e.g. the direct
inference approach and dynamics of soil moisture (Guo and Zhao, 2020). It is unclear whether the predicted results
obtained by one method is justified, which still need further comparisons with the variations in soil water.

In this study, we monitored the isotopic compositions of precipitation, soil, xylem, soil moisture and relevant

variables in *R. pseudoacacia* and *P. tabulaeformis* plantations from 2019–2020 in the Loess Plateau of China. The
objectives of this study were: (1) to investigate the spatial and temporal variations of stable isotopes in consecutive
hydrological years; (2) to quantify seasonal variations of RWU modes for the two main planted species; and (3) to
provide a scientific basis for the optimization of plantations with the combination of soil water storage (SWS).



## 2. Materials and methods

### 2.1 Study site

The experiment was carried out during 2019–2020 in the Caijiachuan catchment on the southeast of the Loess Plateau, China (110°40′–110°48′E, 36°14′–36°18′N), which has an average altitude of 1168 m. The *R. pseudoacacia* and *P. tabulaeformis* plantations were widely planted since implementation of the "Grain for Green" project. The climate is temperate continental, with a mean annual precipitation of 491.6 mm in the period 1985–2020. More than half of the annual precipitation is concentrated from July to September. The annual average potential evapotranspiration is approximately 1723.9 mm. The groundwater table depth is far below 30 m in this area.

The soil is mainly classified as an Alfisol according to the USDA classification system. To weaken the impact of other factors except tree species, we chose sample plots with good growth, similar tree-age and slope aspect, and low human interference. The basic description of the experimental site was shown in Figure 1 and Table 1.

[Figure 1]

[Table 1]

### 2.2 Measurements and sampling

#### 2.2.1 Meteorological measurements

An automatic weather station that monitored air temperature ($T_a$), precipitation (P, TR-525M-R1, Texas Electronics, Inc. USA), relative humidity (RH), solar radiation ($R_a$) and wind speed ($W_s$) at 1 h intervals was located below the hill (with the altitude of 1089 m), which was about 1.5 km away from the study sites. However, because of instrument maintenance, meteorological data from September 9 to November 1 in 2020 was not collected.

#### 2.2.2 Water sampling

Based on the tally method for sample plots of 10×10 $m^2$ of *R. pseudoacacia* (*R.*) and *P. tabulaeformis* (*P.*) forest (Figure 1 (c)), two sample trees were chosen for sampling from each plantation species. During July–August and October, which was regarded as the main growth reason and the end of growth reason, respectively, precipitation,



soil water and stem water were sampled for hydrogen and oxygen stable isotopes analysis.
Two homemade rainfall collectors were used to collect precipitation samples, which consisted of a 500 ml
polyethylene bottle and a 200 ml funnel; a ping-pang ball was loosely placed in the funnel to avoid evaporation.
Precipitation samples were gathered into two 10 ml polyethylene vials after rainfall events (> 2 mm). We selected
and cut suberized branches in the middle and upper canopy with mature bark (0.5–1.0 cm in diameter, 4–5 cm in
length) of *R. pseudoacacia* and *P. tabulaeformis*; xylem water was sampled 1–2 times per month at the same time
of day (13:00–15:00). Bark was quickly removed from the sampled twig, and stored twig in a 50 ml polyethylene
vial with three replicates. Meanwhile, after discarding the first 3 cm of soil to avoid sampling isotopically enriched
water because of evaporation, soil was sampled by auger around the sample tree with three replicates. The fresh soil
sample was divided into two parts, one was used to calculate soil water content (SWC) by the oven drying method
(105℃, 24 h), with 10 (0–20, 20–40, 40–60, 60–80, 80–100, 100–120, 120–140, 140–160, 160–180 and 180–200
cm) and 13 (0–5, 5–10, 10–15, 15–20, 20–40, 40–60, 60–80, 80–100, 100–120, 120–140, 140–160, 160–180 and
180–200 cm) soil layer depths in 2019 and 2020, respectively; the other part was stored in a 50 ml polyethylene
vial. All samples were numbered and sealed with parafilm® and stored in a refrigerator at −4℃ before extraction.
In total, 54 precipitation samples, 54 xylem water samples, and 624 soil water samples were collected.
2.2.3 Isotopic analysis
An automated cryogenic vacuum distillation system (BJJL-2200, Beijing Jianling Technology limited, Beijing,
China) was used to extract soil and stem water in the lab, with cryogenic and heating temperatures of −20℃ and
110℃, respectively; the extraction time and efficiency was 8 h and about 98.0%, respectively. After that, the
extracted water samples and precipitation samples were filtered by needle filter (Polyethersulfone (PES), 0.22 μm,
Membrana company, Germany), and stored in a refrigerator at 3℃ before isotopic analysis. A cavity ring-down
spectroscopy (CRDS) isotopic water analyzer (L2140-i, Picarro Inc. USA) was used for water isotopic
measurements ($\delta^2H$ and $\delta^{18}O$). Results were expressed as parts per thousand relative to the Vienna Standard Mean


Ocean Water (V-SMOW), as in Eq. 1:
$$\delta_{sample} = \left( {R_{sample}} \Big/ {R_{V-SMOW}} - 1 \right) \times 1000‰ \tag{1}$$

where $R_{sample}$ and $R_{V-SMOW}$ are the isotope molar ratios of heavy to light isotopes ($^2$H/$^1$H, $^{18}$O/$^{16}$O) in water
samples and in V-SMOW, respectively. Precision of the CRDS analyzer was ~0.1‰ for $\delta^2$H and ~0.025‰ for $\delta^{18}$O.

To specify the water source of the plant, after the SWC was standardized by z-score, hierarchical cluster

analysis (HCA) method was used to divide the soil layers into four parts based on Ward's method (Figure S1), which
was (1) 0–20 cm; (2) 20–40 cm; (3) 40–100 cm; and (4) 100–200 cm. The details can be seen in Zhao et al. (2018).
The isotope value of each soil layer was calculated by the soil water content weighted mean method
(Mahindawansha et al., 2018), as shown bellows:
$$\delta_{I,J} = {\sum_{i=1}^{n} \sum_{j=1}^{3} \delta_{I,j} * SWC_j} \Big/ {\sum_{i=1}^{n} \sum_{j=1}^{3} SWC_j} \tag{2}$$

where $\delta_{I,J}$ is the representative isotopic compositions of the $I_{th}$ soil layer; $i$ is the number of soil layers in the $I_{th}$
soil layer, and $1 \leqslant n \leqslant 5$; $j$ is the number of samples of the $i_{th}$ soil layer in practice, and $\delta_{I,j}$ and $SWC_j$ is the
corresponding raw isotopic value and soil water content of the sample, respectively.
2.2.4 The direct inference approach and parameter setting in MixSIAR

We assumed that the time delays between sampling and water transport were not significant. The direct

inference approach was applied by directly comparing $\delta^2$H and $\delta^{18}$O between soil and stem. The raw value of $\delta^{18}$O
for stem water and soil water in each interval were applied in this method.

After compared the raw and representative isotope values, the difference between the two values was generally

not significant (Figure S2). Considering the raw data applied in the direct inference approach, the raw isotope values
were also applied in the MixSIAR model for consistent. To determine the contribution of each source to the mixture,
the MixSIAR model, which was based on the Bayesian model principle, was applied to quantify the relative RWU
ratio of different soil water source (0–20, 20–40, 40–100 and 100–200 cm). The mean value and standard deviation
(SD) of isotopic composition of soil water (the source data) and raw plant xylem water (the mixture data) were input



into the model, and the fractionation coefficient was set to zero. The run length was set as "very long" for
convergence, and the model error was evaluated by residual error.
2.2.5 Plant water consumption
Based on the assumption of negligible surface runoff for the two forests, the water consumption of *R.*
*pseudoacacia* and *P. tabulaeformis* plantation was calculated by the water balance equation, as shown bellows:
$$\Delta SWS = \sum \frac{10\Delta (SWC_i d_i)h}{\rho} \qquad (3)$$
$$ET = P - \Delta SWS \qquad (4)$$
where $\Delta SWS$ is the dynamics of layer-cumulated *SWS* in the 0–200 cm soil layers (mm) during a certain time;
$SWC_i$ represents the gravimetric soil moisture (%) in the soil layer; $d_i$ indicates the soil bulk density (g/cm$^3$); $h$
represents the thickness of the soil layer; $\rho$ is the density of water (1 g/cm$^3$); *i* indicates the number of soil layers.
$ET$ is the water consumption per plantations (mm).
2.3 Data analysis
The vapor pressure deficit (VPD) was calculated by the Tetens equation (Gimenez et al., 2019), and the FAO-
56 Penman-Monteith equation was used to estimate the reference evapotranspiration (ET$_0$). Groundwater was not
considered in our study, because groundwater table was well below 30 m in this area and not likely reached by *R.*
*pseudoacacia* and *P. tabulaeformis*.
The mean values of SWC, δ$^{18}$O and δ$^2$H both calculated by the weighted average method. A one-way ANOVA
paired by the least significant difference (LSD) test was performed to identify the differences of the SWC and
isotopic compositions of precipitation, soil and stem water between treatments (*R. pseudoacacia* and *P.*
*tabulaeformis*, drought and humid year). Pearson correlation was conducted to explore correlations between δ$^{18}$O,
precipitation and temperature. All statistical analyses were conducted in R 4.0.3 and MATLAB R2018b.
3. Results
3.1 Temporal variation of environmental variables





Air temperature increased from mid-January to July and decreased from August to January of the next year;
the maximum daily mean temperature was 27.94℃ in July 2019 and 26.34℃ in August 2020 (Figure 2 (a)). There
was 420.0 mm precipitation in 2019 and 605.4 mm in 2020, mainly concentrated in the period July to September;
the precipitation within this period was 242.3 and 371.2 mm in 2019 and 2020, respectively. Meanwhile, the $ET_0$
was 903.17 mm in 2019 and more than 882.36 in 2020, and both highly during June to August (Figure 2 (b)). The
VPD was greater during May to July, which was 0.18–1.75 kPa and 0.20–1.75 kPa in 2019 and 2020, respectively.
Furthermore, according to the Standardized Precipitation Index (SPI, Figure S3) (Mckee et al., 1993), 2019 and
2020 were classified as mild drought and mild humid years, respectively.
[Figure 2]
3.2 Dynamics of soil water and water consumption
Variations in soil water storage (ΔSWS) were calculated for *R. pseudoacacia* and *P. tabulaeformis* forest
(Figure 3). Under the same precipitation, the SWS of *R. pseudoacacia* and *P. tabulaeformis* forest from July to
October increased by 92.42 and 110.60 mm in 2019, and 46.01 and 100.26 mm in 2020, respectively. *R.*
*pseudoacacia* depleted soil water at 100–200 cm from July to August in both 2019 and 2020, while the SWS of 0–
100 cm increased both for *R. pseudoacacia* and *P. tabulaeformis* from August to October in 2019 because of the
increased precipitation during this period. *R. pseudoacacia* and *P. tabulaeformis* also depleted water from the 20–
100 cm and 0–20 cm soil layers, respectively, from August to October in 2020. Both plantations increased SWS
from shallow to deep soil layers over time in 2020. Moreover, the rate of increase in SWS generally decreased with
increasing soil depth in 2019, while the SWS of 40–100 and 100–200 cm soil depths increased more in 2020. For
example, the rates of increases of SWS were 1.10, 0.86, 0.64 and 0.06 for *R. pseudoacacia* forest at soil depths of
0–20, 20–40, 40–100 and 100–200 cm from July to October in 2019, and 0.01, 0.06, 0.14 and 0.21, respectively, in

2020.

[Figure 3]





3.3 Isotopic composition of water samples
With reference to the Global Meteoric Water Line (GMWL, $\delta^2H= 8\delta^{18}O+10$), the relationships between $\delta^2H$
and $\delta^{18}O$ in precipitation and soil water samples were explored (Figure 4). During 2019, the slope and intercept of
the Local Meteoric Water Line (LMWL) were 8.62 and 14.56, respectively, which was higher than those in GMWL,
while these two parameters of LMWL in 2020 were smaller than those of GMWL. Most of the soil water samples
of both *R. pseudoacacia* and *P. tabulaeformis* plantations fell on the bottom of GMWL, and the slope and intercept
of the soil water line (SWL) were also smaller than those in LMWL. The $\delta^{18}O$ values of xylem water of both *R.*
*pseudoacacia* and *P. tabulaeformis* were within the ranges of soil samples.
[Figure 4]
From the perspective of isotopic compositions of water samples (Table S1), the $\delta^{18}O$ of precipitation in 2019
ranged from −8.60 to −6.31‰, with the mean value of −7.20‰ and standard deviation of 0.78‰, while the $\delta^2H$
ranged from −61.88 to −40.41‰, with the mean value of −47.54‰ and standard deviation of 7.23‰. The mean
$\delta^{18}O$ and $\delta^2H$ values of xylem water for *R. pseudoacacia* were −7.99 and −55.50‰, respectively, and −6.76 and
−75.04‰ for *P. tabulaeformis*, respectively. The mean $\delta^{18}O$ value of precipitation, xylem of *R. pseudoacacia* and *P.*
*tabulaeformis* in 2020 was −10.05, −5.29 and −4.82‰, respectively, and −69.43, −67.05 and −66.55‰ for $\delta^2H$,
respectively (Table S2). For total soil water samples of *R. pseudoacacia* in 2020, the mean values of $\delta^{18}O$ and $\delta^2H$
were −10.32±1.81‰ and −72.99±12.12‰, respectively, while they were −8.70±1.33‰ and −66.16±8.16‰ for *P.*
*tabulaeformis* soil water samples, respectively.
The maximum and minimum values of $\delta^{18}O$ and $\delta^2H$ both generally occurred in the 0–20 cm soil layers for
both tree species in 2019 (Table S1), and specifically occurred in the 5–10 cm soil layers in 2020 (Table S2).
Moreover, for both *R. pseudoacacia* and *P. tabulaeformis* plantations, the mean values of $\delta^{18}O$ and $\delta^2H$ of surface
soil layers were relatively high in 2019 and 2020, and water isotopic values were negatively correlated with soil
depth in general, although there was no clear difference in water isotopic values for the soil layers between 100–



200 cm.
3.4 Water uptake pattern based on the direct inference approach and MixSIAR model
Variations of $\delta^{18}O$ in the soil profile and stem during July, August and October in 2019 and 2020 are shown in
Figure 5. With increasing soil depth, the mean value of $\delta^{18}O$ generally first increased and then decreased, and the
soil moisture decreased gradually, except for the SWC in deep soil layers, which was relatively high for *P.*
*tabulaeformis* in July and August 2019.
[Figure 5]
During 2019, *R. pseudoacacia* mainly absorbed water from soil layers of 40–60 and 60–80 cm in July and
August (Figure 5 (a), (b)), respectively, while *P. tabulaeformis* always absorbed water from the 40–60 cm soil layer.
Both tree species absorbed water from 40–60 cm and deeper soil layers in October (Figure 5 (c)), although the SWC
in the 0–60 cm layer increased substantially from August to October. During 2020, water was mainly supplied to
both tree species from the soil layers deeper than 40 cm in July and August (Figure 5 (d), (e)), besides this, water
was absorbed from the surface soil layers by *R. pseudoacacia* in July and by *P. tabulaeformis* in August. Water
absorption was finally concentrated in the soil surface with the continuous increase in SWC (Figure 5 (f)). However,
the method had serious limitations in that it could only roughly determine the soil layers contributing to water
absorption and couldn't quantify the contribution of different soil layers.
Based on the results of MixSIAR model, the contributions of soil water sources from different soil layers to *R.*
*pseudoacacia* and *P. tabulaeformis* varied over time (Figure 6). During 2019, *R. pseudoacacia* evenly absorbed
water from soil layers of 0–20, 20–40, 40–100 and 100–200 cm in July, while *P. tabulaeformis* mainly absorbed
water from 0–20 and 20–40 cm with proportions of 0.37 and 0.36, respectively. The 100–200 cm soil layer
contributed more than half of the water absorbed by *R. pseudoacacia* in August, while *P. tabulaeformis* still mainly
absorbed water from the 0–40 cm soil layer. Lastly, the 40–100 and 100–200 cm soil layers contributed the most to
the both tree species in October. Similar to the results of the direct inference approach, *R. pseudoacacia* and *P.*



*tabulaeformis* both absorbed water from deep to shallow soil depth over time during 2020. The 100–200 cm soil
layer contributed 0.39 and 0.65 to *R. pseudoacacia* and *P. tabulaeformis* in July, respectively, and the 40–100 cm
soil layer contributed the most in August. While the 0–20 cm soil layer contributed substantially to both *R.*
*pseudoacacia* and *P. tabulaeformis* in October, with contribution rates of 0.36 and 0.31, respectively.
[Figure 6]
4. Discussion
4.1 Comparison of the water stable isotope composition of precipitation, soil and xylem

Precipitation was the main input to the regional water cycle, and the water stable isotope compositions reflected

the processes of water vapor transport. The slope and intercept of LMWL were higher than those of GMWL in 2019
(Figure 4), especially the intercept, indicating the larger unbalanced fractionation of water isotopes during the phase
change between vapor and precipitation in 2019. The slope and intercept were 7.65 and 7.48 in 2020, respectively.
This might have been caused by the effect of below-cloud evaporation (Wang et al., 2021b; Xiao et al., 2020). While
the precipitation in 2019 was mainly concentrated in September to October (97.7 mm from July to August, 215.0
mm from September to October), the precipitation samples were mainly sampled from July to August, which were
small and short duration rainfall events (Figure 2 (a)), most likely formed by regional moisture convection (Yamada
and Kurita, 2008; Lynn et al., 1998). The precipitation distribution reversed in 2020 (342.8 mm from July to August,
63.6 mm from September to October), which was highly affected by secondary evaporation, and the high
precipitation resulted in depleted isotope values (Lemma et al., 2020).

The relationship between temperature, precipitation and $\delta^{18}O$ was shown in Table 2.

[Table 2]

Increased temperature and precipitation caused higher and lower isotope values, respectively (Wan et al., 2018;

Dody and Ziv, 2013). This tendency was also demonstrated by the mean value of $\delta^{18}O$, which was lower and
significantly higher than −10.0‰ in 2020 and 2019, respectively (Tables S1 and S2).



From the perspective of the SWL (Figure 4), the soil water isotopic values distributed around LMWL, and the
slope and intercept of both *R. pseudoacacia* and *P. tabulaeformis* plantation were lower than those in LMWL,
indicating that the soil water mainly originated from atmospheric rainfall and intense soil evaporation that occurred
during our study (Liu et al., 2021). Specifically, $\delta^{18}O$ was enriched in the 0–20 cm soil layer in July in both 2019
and 2020 (Figure 5 (a) and (d)), which was caused by the arid climate and rare precipitation prior to this study. This
tendency continued through to August in 2019, while the $\delta^{18}O$ values in the 0–40 cm soil layer became smaller from
July to August in 2020 because the precipitation in 2020 within this period was 2.47 times more than that in 2019.
The new precipitation replenished the soil surface, and pushed the "old" water deeper into the soil profile (Xiang et
al., 2019). This phenomenon was amplified from August to October (Figure 5 (c), (f)). Yang and Fu (2017) proved
that soil water migration in the Loess Plateau was dominated by piston flow, while rare rainwater infiltrated deeper
into the soil profile in the form of preferential flow, and new water (precipitation) evenly mixed with old water (soil
water). The $\delta^{18}O$ values below 100 cm were relatively stable (Figure 5), indicating that they were barely influenced
by precipitation. Furthermore, the vertical distribution of $\delta^{18}O$ in *R. pseudoacacia* plantation was less than that of
*P. tabulaeformis* in 2020, indicating that evaporation under *P. tabulaeformis* was more intense than that under *R.*
*pseudoacacia* during a mild humid year, partially because of the small leaf area of conifer species and relatively
loose stand density (Table 1).
4.2 Comparison of the results of the direct inference approach and MixSIAR model
The direct inference approach suggested that both *R. pseudoacacia* and *P. tabulaeformis* absorbed water from
the 40–100 cm soil layer in July and August, and *P. tabulaeformis* also consumed soil water below 100 cm in October
2019 (Figure 5). While both shallow and deep soil supplied water to plants in July and August, and they mainly
absorbed soil water from 0–20 cm soil depth in October 2020. However, the MixSIAR model pointed that in 2019
(mild drought year), *R. pseudoacacia* absorbed water evenly from different soil layers (0–20, 20–40, 40–100, 100–
200 cm) in July, and absorbed more than half of the total water from the 100–200 cm soil layer in August, while the





depth of absorption shrank to 40–100 cm in October (Figure 6). Using the MixSIAR model, Zhao et al. (2020) also
found that *R. pseudoacacia* preferred deep soil water to support tree growth and nutrient absorption during the dry
season. Our study demonstrated that *P. tabulaeformis* mainly absorbed water at the soil surface (0–40 cm) in July
and August, then extended to soil layers deeper than 100 cm in October; this result was consistent with that of Duan
et al. (2008), who found that *P. tabulaeformis* derived much of its water from rainwater (surface soil) during the
growth season. The two trees had different RWU modes during the drought year, mainly caused by different root
distribution and water availability. Zhou and Shangguan (2006) found that the roots of *P. tabulaeformis* mainly
distributed in the 0–15 cm soil layer and their density decreased with increasing soil depth. However, the root length
and biomass of *R. pseudoacacia* was significantly greater than that of *P. tabulaeformis*, and *R. pseudoacacia* showed
strong drought resistance for its broad and deep root systems (Zhang et al., 2014). *R. pseudoacacia* had a larger
range of soil for water absorption than *P. tabulaeformis*. Moreover, the root vigor gradually decreased with a
decrease in SWC (Li et al., 2011), so the activity of the shallow root system was weakened because of low
precipitation. Thus, *R. pseudoacacia* was able to use deep soil water during the drought year. The trend of water
absorption for both tree species was from deep soil layers (100–200 cm) to shallow soil layers (40–100 cm) to
surface soil (0–20 cm) from July to August to October in 2020 (mild humid year, Figure 6), which was similar to
the results of the direct inference approach. In contrast, the results of the two methods were not completely consistent,
especially during the drought year, which could potentially be explained by the different performance of different
species under diverse environmental stressors.

From the perspective of ΔSWS, the SWS of the 20–100 cm soil layer decreased for the two plantations from

July to August in 2019 (Figure 3), and *R. pseudoacacia* also consumed substantial water from soil layers below 100
cm. This was clearly indicated by the results of the MixSIAR model (Figure 6), where the contribution rates of the
100–200 cm soil layers were 0.24 and 0.52 in July and August, respectively. Moreover, the results of MixSIAR were
more specific and coordinated with the dynamics of SWS in 2020, for example, the contribution rate of the 20–100

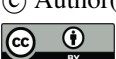



cm soil layer to *R. pseudoacacia* was considerable in October, while the direct inference approach manifested the
surface soil only, although it indicated the upward trend of the main soil layer supplying water to *R. pseudoacacia*
from July to October (Figure 5). Tetzlaff et al. (2021) also found that the isotopic values of xylem water were usually
dissimilar to those of soil water in the drier months, which weakened the applicability of the direct inference
approach. In general, the MixSIAR model was more accurate and applicable in the study of water uptake mode for
this region.

To explore the relationship between SWC and RWU, a correlation analysis was conducted between SWC and

contribution rates (based on the MixSIAR model) of the 0–20, 20–40, 40–100 and 100–200 cm soil layers. The
results showed that the correlation between the two was not significant, indicating that the process of RWU was a
combination of many factors, including SWC, root distribution, soil properties and meteorological factors (Zhao et
al., 2020). While there was a tendency that the increasement of SWC facilitated the contribution rates for *R.*
*pseudoacacia*, as shown below:
$$\Delta CR = 1.07 \times \Delta SWC - 0.33 \qquad R^2 = 0.73 \qquad\qquad (4)$$
where $\Delta CR$ and $\Delta SWC$ were the degree of change in the contribution rates (CR) and SWC, respectively. While
this trend was not clear for *P. tabulaeformis*.
4.3 Plant water consumption and management

The SWS of the 0–40 cm soil layer from August to October in 2019, the 20–40 cm soil layer from July to

August and the 40–200 cm soil layer from August to October in 2020 showed the greatest increase (Figure 3),
meaning that water was only restored in shallower soil during the mild drought year, while it continuously infiltrated
to deeper soil during the mild humid year. In total, the SWS of *P. tabulaeformis* plantation increased more than that
of *R. pseudoacacia* from July to October, especially during the mild humid year (in 2020), indicating that *R.*
*pseudoacacia* consumed more water than *P. tabulaeformis* after neglecting the differences in surface runoff and soil
evaporation (Jian et al., 2015; Shen et al., 2020). Both species consumed more water in the humid year than in the



drought year, demonstrated by the similar SWS at the end of the growth season under increased precipitation from
2019 to 2020.
When combined with the results of water apportionment, *R. pseudoacacia* water absorption extended to deeper
soil layers when the water supply was insufficient, while *P. tabulaeformis* water absorption was restricted to the
surface soil (0–40 cm). With the supply of large precipitation volumes, the main soil layers of RWU for *R.*
*pseudoacacia* shrank to surface. However, both species absorbed water from deep to shallow soil layers with
continuous sufficient water supply by rainfall through to the end of growth season. In summary, the water use
strategy of *P. tabulaeformis* was conservative, while it was more alterable and hydrotactic for *R. pseudoacacia*.
Our study found that the reference evapotranspiration was greater than annual precipitation in this region.
However, countless trees with high density were planted during the afforestation programs, without considering the
limited precipitation (Jia et al., 2017). Increasing canopy transpiration and interception both highly weakened the
soil water storage, the plants with deeper root systems were promoted to absorb water from deep soil layers to
alleviate water stress. With the characteristics of well-developed roots, *R. pseudoacacia* had a higher resilience
under drought. Furthermore, *R. pseudoacacia* consumed more water throughout the growing season, especially from
deep soil layers (100–200 cm), causing potential threat for soil desiccation, and led to imbalanced water cycle and
ecological degradation. Compare to the *R. pseudoacacia*, *P. tabulaeformis* consumed less water than *R.*
*pseudoacacia*. However, *P. tabulaeformis* preferred to absorb water from shallow soil layers, which growth may be
limited by precipitation. The continuous absorption from shallow soil would plunder the water sources from
understory vegetations, and may not conducive to the construction of functional ecological forest.
Considering the regional water conservation and water cycle, the artificial *P. tabulaeformis* plantation was
better than the *R. pseudoacacia*. However, a combination of different plant species with different water use strategies
would form a good community, which would also good for the long-term and sustainable development of forest
ecosystems. And reasonable thinning should be applied for artificial plantations, which needs further detailed





research.
5. Conclusions
In this study, we explored the spatial and temporal variations of water stable isotopes of precipitation, soil
water and xylem, and we investigated the water uptake patterns of *R. pseudoacacia* and *P. tabulaeformis* by means
of the direct inference approach and the MixSIAR model, and taking the variations in SWS as a reference. The $\delta^{18}O$
values of precipitation were positively and significantly negatively correlated with temperature and precipitation
volume, respectively. The isotopic compositions of surface soil water varied with seasons, while those of the soil
layers below 100 cm were relatively stable.
Compared with the direct inference approach, the MixSIAR model performed better in quantifying dynamics
of RWU modes; and was consistent with the variations in SWS. The model results showed that *R. pseudoacacia*
and *P. tabulaeformis* had different RWU modes, especially in a drought year. *R. pseudoacacia* mainly absorbed
water from the 100–200 cm soil layer in the drought season, and *R. pseudoacacia* showed strong drought resilience
with the flexible water use strategies, while *P. tabulaeformis* consumed water mainly from the 0–40 cm soil layer
under drought. However, the water absorption for both tree species changed from deep soil (100–200 cm) to shallow
soil (0–40 cm) with continuous water input during the humid year. Furthermore, *R. pseudoacacia* consumed more
water than *P. tabulaeformis,* especially in the mild humid year, and preferred to extract deep soil water in the drought
year, which could induce soil desiccation and be harmful to the sustainable development of forest ecosystems. This
study evaluated water use characteristics of two mainly planted trees in Chinese Loess Plateau, and provides
scientific basis for plant species selection and forest restoration and management.
***Code and Data availability***. Codes applied in the MixSIAR model are available upon request to the authors. Isotopic
data and data of soil water are available upon request to the authors.
***Author contributions***. Chengzhong Pan, Yongsheng Cui and Lan Ma designed the research. Yongsheng Cui and
Zhanwei Sun conducted the experiment and collected the data. Zhanwei Sun made some data curation, Yongsheng



Cui analyzed the data and wrote the original manuscript. Chengzhong Pan and Lan Ma provided suggestions to the
original manuscript.
***Conflict of interest***. The authors declare that they have no conflict of interest.
***Acknowledgements.*** This research was financially supported by the National Natural Science Foundation of China
(Grants 42077059 and 41771305).

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





Tables
Table 1    Basic description of the study sites

| Site | Main plant type | Slope gradient (°) | Slope aspects (°) | Tree-age (yr) | Tree-height (m) | DBH (cm) | Stand density | Soil (0–200 cm) Bulk density (g/cm³) | Field capacity (g/g) |
|------|-----------------|--------------------|-------------------|---------------|------------------|----------|---------------|------------------------|----------------------|
| *R.* forestland | *Robinia pseudoacacia, Rosa xanthina, Artemisia sacrorum, Carex spp.* | 22 | 283 | 26 | 8.73±0.77 | 10.20±0.47 | 2×3.5 m | 1.27 | 0.42 |
| *P.* forestland | *Pinus tabulaeformis, Carex spp.* | 23 | 315 | 29 | 6.73±0.42 | 11.06±0.43 | 2.5×4 m | 1.25 | 0.42 |

NOTE: DBH means the diameter of trees at 1.3 m above the ground. Values of the mean ± SD were presented.



Table 2    Relationships between temperature ($T_a$), precipitation volume ($P$) and $\delta^{18}$O

| Year | 2019 | | 2020 | |
|------|------|--|------|--|
| $T_a$-$\delta^{18}$O | $\delta^{18}O = 0.28T_a - 12.97$ | $R^2 = 0.25$ | $\delta^{18}O = 0.21T_a - 12.88$ | $R^2 = 0.43\,*$ |
| $P$-$\delta^{18}$O | $\delta^{18}O = -0.11P - 6.03$ | $R^2 = 0.76\,*$ | $\delta^{18}O = -0.22P - 7.65$ | $R^2 = 0.54\,**$ |

NOTE: * and ** means P<0.05 and P<0.01, respectively.







Figures


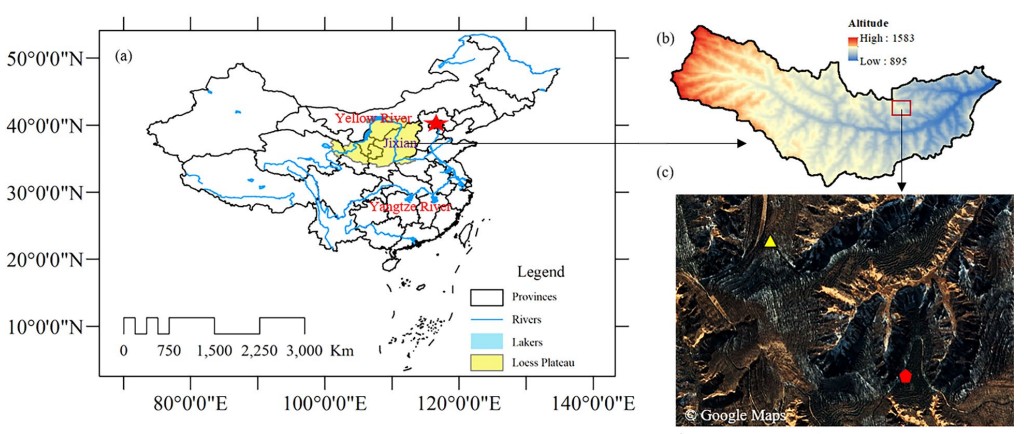


Figure 1 The study sites in Chinese Loess Plateau (a), and the Caijiachuan catchment (b). (c) google map of the
experimental sites for *Robinia pseudoacacia* (*R.*) and *Pinus tabulaeformis* (*P.*) plantations.















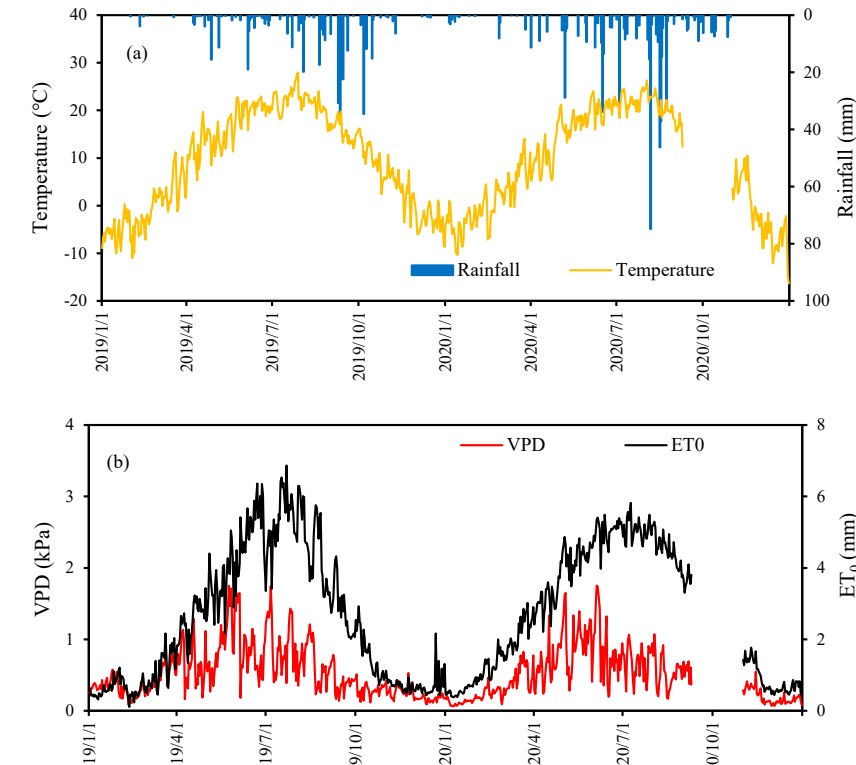



Figure 2    Daily dynamics of temperature, rainfall, vapor pressure deficit (VPD) and reference evapotranspiration
($ET_0$) in 2019 and 2020.












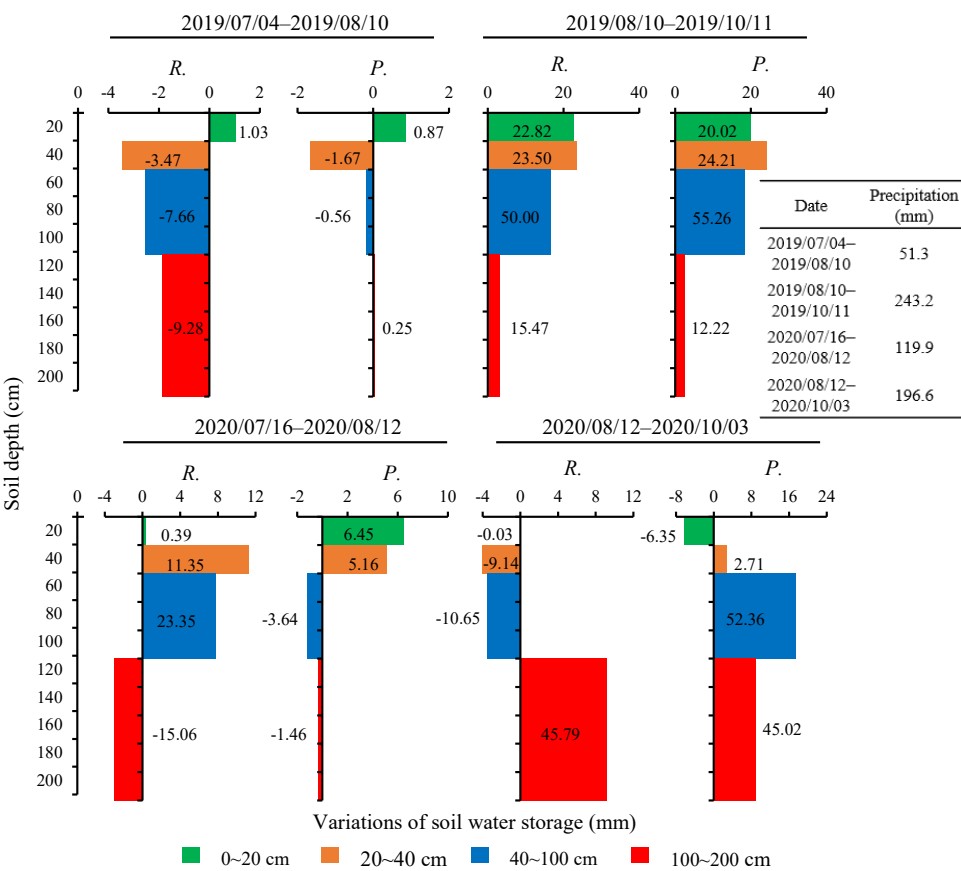


Figure 3    Variations of soil water storage (the area of the blocks) along the soil depths during our study (from July to October) for *R. pseudoacacia* (*R.*) and *P. tabulaeformis* (*P.*) plantation, and the number noted in the figure was the variations of soil water storage of different soil layers (0–20, 20–40, 40–100, 100–200 cm) at the corresponding time. The precipitation at the corresponding time was shown in the table.










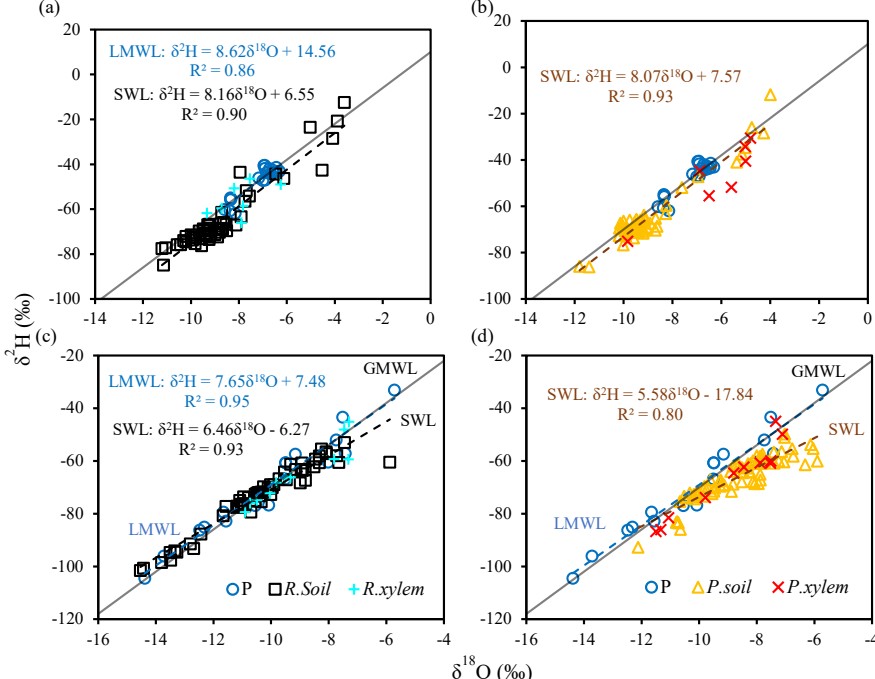

Figure 4    Relationship between $\delta^2$H and $\delta^{18}$O in precipitation (LMWL), soil water (SWL) and xylem water. (a)
and (b) was *R. pseudoacacia* and *P. tabulaeformis* forest in 2019, respectively; (c) and (d) was *R. pseudoacacia* and
*P. tabulaeformis* forest in 2020, respectively. And the grey solid line represents the Global Meteoric Water Line
(GMWL), the blue, black and orange dashed line represents the LMWL, SWL of *R. pseudoacacia* and *P.*
*tabulaeformis*, respectively.











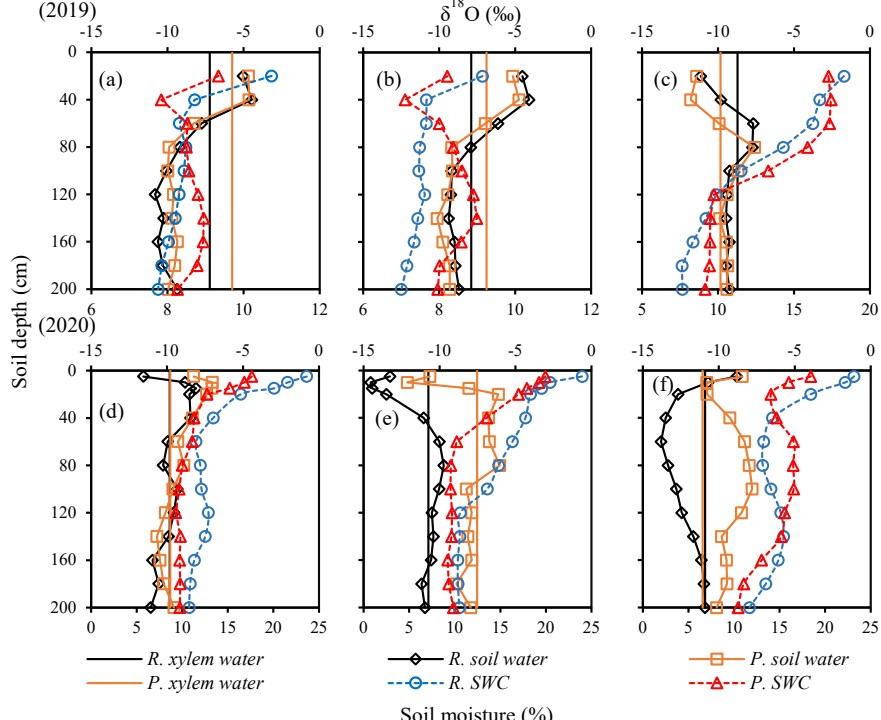


Figure 5    Mean values of δ<sup>18</sup>O in xylem and soil water along the soil profile, and corresponding soil moisture for
*R. pseudoacacia* and *P. tabulaeformis* forest. (a) and (d), (b) and (e), (c) and (f) was the samples in July, August,
October in 2019 and 2020, respectively.












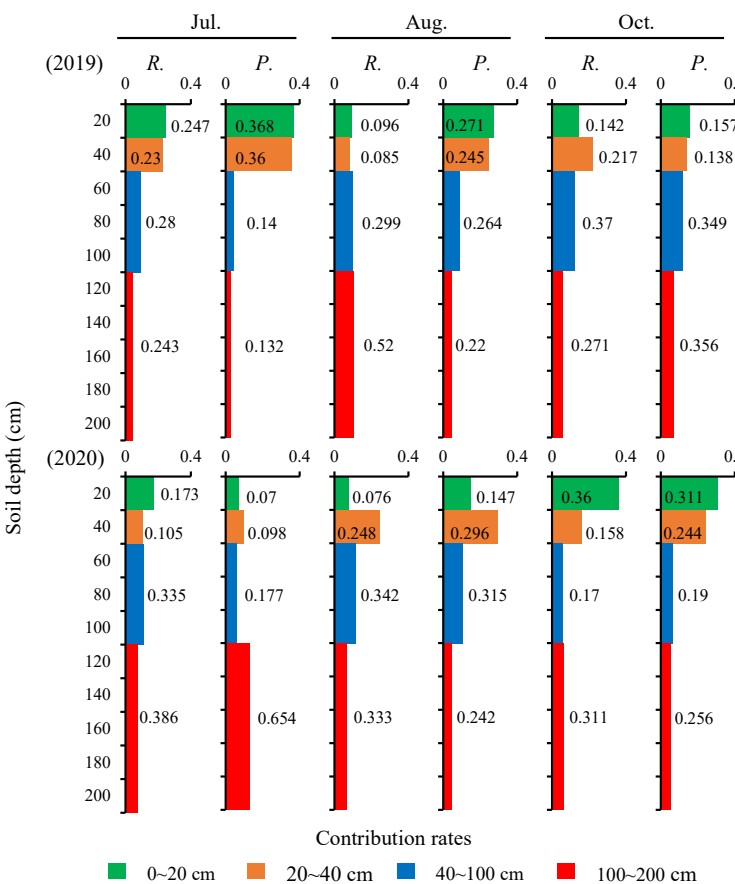


Figure 6    Contribution rates of soil water (the area of the blocks) to *R. pseudoacacia* (*R.*) and *P. tabulaeformis* (*P.*)
based on the MixSIAR model, according to the values of $\delta^{18}O$. The number noted in the figure was the contribution
ratio of the corresponding soil layers (0–20, 20–40, 40–100, 100–200 cm).