# Peer review of "Comparing water uptake patterns of two plantations using stable isotopes in Chinese Loess Plateau"

_Hydrology and Earth System Sciences, 2022_

## Referee Comment (RC1)

**Review for HESS-2022-120: Comparing water uptake patterns of two plantations using stable isotopes in Chinese Loess Plateau**

Dear editors, dear authors,

After careful consideration of the submitted manuscript I recommend that the presented analysis needs to undergo numerous and extensive revisions (see more detailed comments below). At the same time, I encourage the authors to refine the analysis and re-submit an improved manuscript to present this novel data set. I hope that the remarks in this document help them to do so in an efficient manner.

- The paper addresses the vertical patterns of depth of root water uptakes (RWU) by two tree species located in the Chinese Loess plateau. It investigates how these RWU patterns change over the season and also comparing a dry with a wet year. The paper sets out to compare RWU obtained with different methods: MixSIAR, Direct Inference (both based on isotopic signatures in soil and xylem waters) and a method based on the change in soil water storage. This comparison is most welcome in the research community and lies in the scope of HESS. However, tackling this requires in-depth discussion of the assumptions underlying these methods and the corresponding limitations.
- The researchers collected an excellent data set with many replicates that allows to have robust conclusions. The data set is novel to my knowledge and should definitely be published! This involved a lot of field work and has value in it. It is great to see, that the aggregated values are reported in the supplementary material. (Although, raw values would be even better.)
- Scientific methods and assumptions are not clearly stated and the limitations of these methods could affect the interpretations and conclusions. The analysis is subject to ambiguities and lacks clear figures, which leads to the provided materials not robustly supporting the claims and findings the authors make in the discussion (e.g. confounding factor P in ΔSWS, or figure 6 not clearly supporting l.353-354, among others). Improvements to the analysis and illustration of the presented data are needed in order to assess how well they support the claimed findings.

**General comments**

- The article is clearly structured, but suffers from imprecise language and formulae. This hampers its clarity and reproducibility. Further, many figures should be tweaked to better support the comparisons and care should be taken to the consistent reporting of units.
- The results and discussion of the Meteoric Water Lines (slopes and intercepts) is not adding significantly to the outlined research questions or discussed findings and could be removed. An exception to this is the visible evaporative enrichment in soil and xylem water that highlights differences between the dry and humid year and the fact that the dual isotope plots indicates possible source waters of xylem seem to be sampled exhaustively.
- In order to interpret ΔSWS, one has to notice that it is a combined effect of multiple processes, which can be simplified to ΔSWS = P - ET. Thus, precipitation P is a confounding factor that can influence estimates of evapotranspiration ET based on ΔSWS. This confounding factor P has differing impact given the amount and duration since last reasonably large rainfall when ΔSWS is measured. This severely limits the possible comparisons that can be made based on ΔSWS. For example, P is the same when the authors compare the same seasonal period between P. tabulaeformis and R. pseudoacacia.

However, it is not the same e.g. when the authors compare across years or across seasonal periods. This should be stated more explicitly and authors should either a) limit their comparisons to the ones unaffected by this, b) clearly discuss the impact, or c) try to remove the effect of P in ΔSWS to allow comparison of ET. (This affects various lines in the manuscript, e.g. l.381-382 in the discussion.)

- The authors mention the drought resilience of R.pseudoacacia based on the species' ability to increase the relative contribution from deeper soil layers to RWU in drought years (l.39-41). What would additionally be interesting is to see if this resilience allows to maintain a comparable total amount of transpired/absorbed water in dry vs wet years (e.g. ET = ΔSWS - P). I would be interested by an additional, more detailed analysis combining the mixSIAR based RWU (relative contributions) with the total amounts of actual ET (derived as ET = ΔSWS - P, assuming no deep percolation as a first approximation), to estimate if total contribution from 40-200cm was increased in absolute terms in 2019 compared to 2020 (l.39-41).

**Specific comments**

- To clarify figures and make them better support the results please consider:
  - make x-axes comparable by having same limits (concerns Fig.3, Fig.4, Fig5 (SWC))
  - adding title and units on x axes (Fig.3, Fig.6) (Fig.3 How is the x-axis in Fig 3 linked to SWC?)
  - consider a horizontal layout with a row for R.pseudoacacia and another row for P.tabulaeformis, (agreeing with the suggested table below) (concerns Fig.3 and Fig.6)
  - For figure 6: I feel that using the area to represent the total relative contribution of each layer makes it difficult to compare their values. I argue that using bar width to represent total relative contribution (in combination with a constant height of the rectangles) would better support your discussion of total relative contributions. (Indeed you're not discussing relative contribution per unit depth, which is currently shown by the rectangles' widths.)
- Revise language (especially tenses in figure descriptions but also correct conjugation of verbs)
- Be careful to remain consistent in your terminology. E.g. refer consistently to "twig xylem water" instead of "stem water".

Find further specific comments pertaining to the indicated lines below:

**Intro**

l.61-62: Please clarify what you mean by "depict all water sources". Further it'd be nice to see a discussion on the limitations, e.g. the confounding factor of P that limits the interpretation of RWU=ET=P-ΔSWS. And what comparisons are allowed or not considering this confounding factor.

l.70-73 and l.79-81: Please try to be more concise and clear in the English. For the reader it is not easy to understand the message of these lines.

**Materials and Methods**

l. 118: How was the number of two trees per plantation decided? Does this mean you took multiple twigs then from the same tree? Were the results between the two trees consistent? This could be added as a small sentence to the results.

l.148: how are the four "aggregate" layers linked to the HCA shown in figure S1? It would be nice to have some more background information on how you applied Ward's method (applied to what variable? Z-scores?)

l.150-155: Does "I" refer to the "aggregate soil layer" (0-20, 20-40, 40-100, 100-200cm?) while "i" is the sample depth (i.e. the ones state in lines 130-131) ? Maybe revise the English to make this distinction clearer in the text.

l.164-167: Pleaser provide more details to the MixSIAR model, e.g. did you use any prior assumption on the root distribution to constrain solving the overparametrized system? How did these root distributions look like?

l.170-176 Please revise the formula a) generally (units, explain $\Delta$) and also b) in view of the comments to lines 150-155. Unclear points include (should h be $h_i$? Maybe include the double index I to explain how you computed quantities shown in figure 3? Was $d_i$ constant over time or did you really consider $\Delta d$ as your formula suggests with the parentheses?)

l.185-186: Please provide more explanations why you look at Pearson correlation between variables (also affects l.233-234, 276-280, Table 2, and l.333-342). To understand l.333-342 a better explanation of the hypotheses underlying the correlation analysis between $\Delta SWC$ and $\Delta RWU$. It would also be nice (on line.185-186) to have further details on the subgroups within wich you computed correlations. These analyses could lead to a better process understanding of the water cycle but could be more clearly linked to the research question regarding forest restoration schemes or more clearly linked to the interpretation of the other analyses.

**Results**

l.201 Please mention that the reported values in the text are "(not shown)" in the figure 3 as such. Further you might add that in 2019 as well as in 2020 total changes in SWS between August and October were much bigger than changes in SWS between July and August.

l.201-210: This paragraph could greatly benefit from more clarity by reworking the text (both structure and language) to make it easier to follow the various comparisons. A support in form of a table or figure would definitely be helpful. Additionally the confounding factor/ambiguity induced by P should be more highlighted for the respective comparisons. E.g. would the time since last rainfall affect the observed decrease/increase in $\Delta SWS$ (or "rates of increase of SWS") with depth or is the observed $\Delta SWS$ robust with respect to that because you consider depths of up to 2m ? A discussion of this would be needed.

l.213-219: Please explain better the relevance of reporting the slope and intercepts of the LMWL and SWL, as well as the ranges of the observed isotope values (l.221-230). Where they do not contribute to the findings regarding the research question, they could be removed (see general comment further up.) This also applies to correlation with soil water (l.232-234)

l.242-261: These result sections would benefit from a summary in tabular form, where you might compare RWU depths from the direct inference method, RWU modes from the MixSIAR method and largest $\Delta SWS$ for the 6 periods and the two plantation types. See below example as suggestion (values to be verified...):

| | Dry year | | | Humid year | | |
|---|---|---|---|---|---|---|
| Plantation species | Jul 19 | Aug 19 | Oct 19 | Jul 20 | Aug 20 | Oct 20 |
| **Direct inference**: RWU depth (cm) | | | | | | |
| R (black) | 60? | 80? | 50 | 10,70,150? | 40? | 20? |
| P (orange) | 50 | 60 | 70 | 120? | 20? | 20? |
| **MixSIAR model**: Depth of RWU mode (cm) | | | | | | |
| R.pseudoacacia | 40-100 | 100-200 | 40-100 | 100-200 | 40-100 | 0-20 |
| P.tabulaeformis | 0-20 | 0-20 | 100-200 | 100-200 | 40-100 | 0-20 |
| **Change in SWS**: Depth of largest decrease in ΔSWS (cm) | | | | | | |
| R.pseudoacacia | ... | | | | | |
| P.tabulaeformis | ... | | | | | |

**Discussion**

l. 331-332: Also here above table could help clarify when and where MixSIAR agrees with ΔSWS and when and where it does not. It appears to me the conclusion that they agree well is mostly based on the observation of water use from 100-200cm in R.pseudoacacia (in July and August 2019). See also remark regarding l.353-354. l.322-324

l. 333-340: Please explain better the hypotheses underlying the correlation analysis between ΔSWC and ΔRWU and what you want to test with it. E.g could the same have been done instead with ΔSWS (ΔSWS = \bar{ΔSWC} * 10 *\bar{d} * h / ρ) ?

l.353-354: How is this claim supported by the MixSIAR figure 6 that implies 65% of xylem water coming from 100-200cm for P in July 2020 or 26% for P in October 2020 ? Is this claim only based on figure 3 based on ΔSWS method? (l. 331-332)

**Conclusions**

l.378-380: as stated earlier it is still unclear why correlation of δ18O values with other variables is analysed and (why this is stated as such in the conclusion).

**Technical corrections**

**Abstract**

l.34: suggest to remove "with contrasting soil moisture dynamics"

l.35-36: what does correlation of δ with Tair and P mean? How to interpret this?

l.38: "more effectively" what does it mean?

l.43: "inclined to absorb soil layer" what does it mean?

**Intro**

l.53: replace "potential" with "danger"?

l.92: Please try to avoid confusion in the English formulations e.g. distinguish "soil δ" from "soil moisture content" and be more explicit in the "relevant variables".

l.94: By "spatial" you mean "vertical"? Would this be a more appropriate formulation?

l.101: Refer to a year instead of the "Grain for Green" project.

**Materials and Methods**

l.138: cryogenic extraction: could you report the efficiency separately for soil and twig samples?

l.146: how did you measure the precision of the CRDS analyzer? Do you have an estimate of the accuracy?

l.161: with "significant" do you mean "relevant" or "large"?

l.157-165: please clarify at the very beginning of the paragraph what data set was used for the analysis and all the figures (except Fig S2). I understand it was the "raw" (and supp-l.10 do you mean "unweighted average method")

**Results**

l. 205-210: "Both plantations increased SWS ... " is unclear? Is this saying that ΔSWS from July-October for each single depth and for both years was positive? Please add units to the "rates of increase" stated in the text and consider showing/adding these numbers in tabular form to follow more easily the claims.

l.242-244: State more clearly that these are results from direct inference approach.

**Discussion**

l.267: Do you mean "kinetic fractionation" (as opposed to "equilibrium fractionation") instead of "unbalanced fractionation"?

l. 290: Consider replacing "rare rainwater infiltrated" with "on rare occasions rainwater infiltrated".

l. 333: Consider adding "[the relationship between] the changes in [SWC and RWU]".

l.357: Please explain what you mean by conservative.

l.359: Consider replacing "trees with high density were planted" by "trees were densely planted"

l.365: "Compare to the R. pseudoacacia" is redundant at the beginning of the sentence.

l.370: "better" in the sense to consume less water and allow more deep infiltration?

**Figures**

l. 531: Consider replacing "google map" with "Satellite view (Google Maps)"

l.577: Consider clarifying caption: "Grey and blue lines represent the Global (GMWL [add formula]) and Local (LMWL) Meteoric Water Lines, black and orange represent the Soil Water Line of the R... and P... sites, respectively."

l.588: Consider adding "[in xylem] (shown as vertical lines) [and soil water]"

---

## Author Comment (AC3)

**RC2**

Comparing water uptake patterns of two plantations using stable isotopes in Chinese Loess Plateau

Comments to the authors

The topic treated in this manuscript seems to be of great interest for the studied regional area, namely the Loess Plateau of China. As described in detail by the authors, an evaluation of the RWU of different tree species is pivotal to better understand the water consumption related to the presence of artificial tree plantations in this area. In particular, this is pivotal to shed light on the consequences of massive afforestation and how they affect the delicate balance of an ecosystem. Therefore, in my opinion, the study fits the need for scientific data to determine the sustainability of agricultural practices, in terms of guaranteeing the sustainable development of the Loess Plateau and mitigating water scarcity.

**Authors response:**

Thank you for your affirmation of our study. The massive afforestation greatly changed local hydrological processes, caused decreased river runoff and increased danger of soil desiccation. It is important to understand how vegetations affect the local water balance and the adaption of vegetations to different water conditions, which would provide scientific basis for forest management and sustainable development of forest ecosystems.

The study compares two sites characterized by different tree species, R. pseudoacacia and P. tabulaeformis. Tree and soil samples were collected periodically within two growing seasons to determine the soil water content and water sample isotope composition. To establish the soil depth at which RWU occurs, the results of the direct inference method were compared to the results of the MixSIAR model. Moreover, meteorological data (precipitations) and the soil water content are combined to estimate the overall evapotranspiration for the site. In my opinion, the study is well described and contextualized, however, improvements are needed to make the text more effective and comprehensible to readers. In particular, the M&M section lacks more precise information about the structure of the sampling, which is unclear in some parts (how many trees were selected per site? Different trees at each sampling? How were they selected? An overall description of the principal characteristics of the two tree species in each site would also be needed). Obviously, the decision of the sampling size in an extensive sampling campaign has to take into account several factors (time, budget, material, etc.), but considering the large variability that could be found in natural environments, the choice of two samples per site per sampling seems a rather limiting factor to result interpretation. Moreover, I believe the authors should have included additional measurements of the tree physiological conditions (have you measured any difference in the water stored by plants in mild dry and mild humid seasons?) and an in-situ investigation of the tree root system, to further support the results of the model. I invite the authors to consider these points in further studies. For what concerns this manuscript, I would suggest some improvements and revisions to the text before publication. Additionally, I invite the authors to carefully revise the English usage to improve the text clarity, conciseness, and wording.

**Authors response:**

Thank you for your warm and constructive comments. More detailed and precise information will be added in the "M&M" in the revised manuscript. After the tally of sample plot, two sample trees with the mean tree-height and DBH (Table 1) were selected and sampled throughout the experimentation per site. The basic overall description of the two plantations can be seen in Table 1. Although only two trees were selected for each plantation, the physiological parameters of the two trees were representative for per plantation and tree species. And we conducted long-term continuous fixed observation, which makes our research results more credible.

Moreover, the plant water storage was measured by other research teams, and the trunk stem water content of *Robinia pseudoacacia* and *Pinus tabulaeformis* was 12.5% ad 14.8%, which was generally the same with the similar tree-height (Table 1). The vertical distribution of plant roots was also studied in a similar plantation by other research

teams, which we could applied in our study. The discussion of plant water storage and roots distribution will be added in the revised manuscript. Furthermore, we will revise our English usage and double-check our statement in the revised manuscript. Thanks again for your comments, the point-by-point responses were listed as shown below.

**Hereafter, a list of points that require adjustments before publication.**

Line 51: "Soil moisture is the main water source of vegetation growth". Revise the use of the prepositions.
**Authors response:**

Thank you for your detailed comments. We will replace "of" with "during" in the revised manuscript.

Line 60: "as an active layers" please correct.
**Authors response:**

Thank you for your detailed comments. "as an active layers" means the SWC of 0–200 cm soil layers was the main water source for plants and easily influenced by precipitation. We will rewrite the sentence for clarify in the revised manuscript.

Line 65: What do you mean with water transformation? Do you refer to the physical changes of the water molecules?
**Authors response:**

Thank you for your detailed comments. We are sorry for the inaccurate formulation, we will replace "water transformation" with "water transition" in the revised manuscript.

Line 73-76: the sentence should be revised, as it is not clear.
**Authors response:**

Thank you for your constructive comments. We will rewrite the sentences for clarify in the revised manuscript.

Line 74: As you refer to a comparison among different species, which species were selected?
**Authors response:**

Thank you for your detailed comments. The four species were *Artemisia capillaris*, *A. sacrorum*, *Bothriochloa ischaemum* and *Lespedeza davurica*, we will rewrite the sentences in the revised manuscript as we have stated above.

Line 79: "These studies mainly focused on plants with yearly leaf abscission and studied the seasonal variations generally". But a study in which R. pseudoacacia was studied is just mentioned, and only two other plant species are reported in this section.
**Authors response:**

Thank you for your constructive comments. We will add more recent researches to enrich this section (Lines 70-83) in the revised manuscript.

Line 91: "It is unclear whether the predicted results obtained by one method is justified" Which method?
**Authors response:**

Thank you for your detailed comments. "one method" means one of the three methods: the direct inference approach, the MixSIAR model and the dynamics of SWS. One or two methods were widely used in the RWU investigation, while the comparison between the three methods were relatively rare. We will rewrite the sentence for clarify in the revised manuscript.

Line 93: "from 2019–2020" Revise the use of the prepositions.
**Authors response:**

Thank you for your detailed comments. We will replace "from 2019–2020" with "in 2019–2020" in the revised manuscript.

Line 96: "3) to provide a scientific basis for the optimization of plantations with the combination of soil water storage (SWS)". Not clear. Please revise.
**Authors response:**

Thank you for your constructive comments. The third objective of our study was to compare the water use strategies of the two species, then provide scientific basis for forest management with the consideration of water consumption. We will rewrite this objective for clarify in the revised manuscript.

Line 100: "The R. pseudoacaciaand P. tabulaeformisplantations were widely planted since implementation of the "Grain for Green" project". Additional information related to the two species? Age, height, dimension of the crown, diameter, leaf area?
**Authors response:**

Thank you for your detailed comments. The basic description of the two plantations was shown in Table 1. The specific physiological parameters for the representative trees, including age, height, dimension of the crown, diameter, will be added in the revised manuscript.

Line 101-107: Can you provide a reference for the information in this section?
**Authors response:**

Thank you for your detailed comments. We will add a reference for the information of the study site, e.g., 10.1016/j.agrformet.2022.108908.

Line 118: Did you use only two trees for the whole sampling campaign, conducted in two consecutive years with periodical tree and soil sampling? Or were two different trees within the same plot at every sampling selected? How were the trees selected? In any case, I believe that the number of trees is rather low for such a study, limiting data interpretation. How do you justify this choice?
**Authors response:**

Thank you for your warm and detailed comments. *Robinia pseudoacacia* and *Pinus tabulaeformis* were the main planted tree species, accounted for more than 30% of the afforestation area in Chinese Loess Plateau, which were also widely planted in the study site. After the tally of the sample plots ($10 \times 10$ m$^2$) in the plantation (Figure 1 (c)), two representative trees, located in the central of the plantation, were selected by the mean DBH and tree-height (Table 1) per plantation, and the sampling was conducted on the two periodical representative trees in two consecutive years. The physiological parameters of the two trees were representative for the whole plantation, and the RWU modes of these two trees are also representative of the species.

Line 119: "which was regarded as the main growth reason and the end of growth reason". Season instead of reason?
**Authors response:**

Thank you for your detailed comments. We apologized for our superficial negligence, and we will replace "reason" by "season" in the revised manuscript.

Line 122: "ping-pang ball" → ping-pong ball.
**Authors response:**

Thank you for your detailed comments. We will replace "ping-pang ball" by "ping-pong ball" in the revised manuscript.

Line 124: "in the middle and upper canopy". Can you specify the middle and upper height of the canopy? Which is the total height of the trees?

**Authors response:**

Thank you for your detailed comments. The middle and upper canopy of *Robinia pseudoacacia* is at the height of 6.0–7.5 m and 7.5–9.0 m, respectively, and it is 4.0–5.5 m and 5.5–7.0 m of *Pinus tabuliformis*, respectively. The height of the two representative trees was 8.83 m and 8.94 m for *R. pseudoacacia* plantation, and 6.90 m and 6.75 m for *P. tabuliformis* plantation. We will add detailed information in the Table 1 and Lines 117-118 in the revised manuscript.

Line 126: "and stored twig" → and twigs were stored

**Authors response:**

Thank you for your detailed comments. We will rewrite the sentence and replace "and stored twig" by "and twigs were stored" in the revised manuscript.

Line 128: "soil was sampled by auger around the sample tree with three replicates". Was the soil collected every time around the same trees? As pointed out in my previous comment, it is not clear if the trees for the sampling were always the same or different at each sampling. At which distance from the tree was the soil collected?

**Authors response:**

Thank you for your detailed comments. The soil samples were collected around the two representative trees every time, and the two representative trees were sampled periodically. The distance from the representative tree to soil sampling point was about 50–150 cm. We will add more detailed information in the revised manuscript.

Line 132:" the other part was stored in a 50 ml polyethylene vial" for which purpose? Isotope analysis? Please be more precise.

**Authors response:**

Thank you for your detailed comments. The soil samples stored in the vial was for isotope analysis. We will add more detailed information for clarify in the revised manuscript.

Line 157: "We assumed that the time delays between sampling and water transport were not significant". Do you mean the difference in the isotope composition of the soil and the tree samples, due to the time delay between root water uptake and water distribution in the twigs? Please revise the sentence.

**Authors response:**

Thank you for your detailed comments. Plants absorb soil water by roots then transmit it into twigs, we assumed that there is no time lag between RWU and plant transpiration. The water isotope composition of the soil and the twigs were the same at the same time. We will rewrite the sentence: "We assumed that time delays between RWU and water transport in the twigs were not significant, and the water isotope compositions of the soil and the twigs were generally the same at the similar time".

Line 160: "After compared the raw and representative isotope values," → after comparing

**Authors response:**

Thank you for your detailed comments. We will replace "compared" with "comparing" in the revised manuscript.

Line 165: "raw plant xylem water". In line 79 the authors introduce the terms plant and tree, underlying the difference between the two, particularly in relation to studies of the RWU. It is then added that this study is focused on artificially planted trees. To be more consistent throughout the text, I suggest using the same terminology. Please

check the text for incongruities.

**Authors response:**

Thank you for your detailed comments. The term "tree" is included in the "plant" in our opinion, we will replace "tree xylem water" with "plant xylem water" for consistent in the revised manuscript.

Line 173-176: Please, specify the meaning of P in formula 4.

**Authors response:**

Thank you for your detailed comments. "P" means precipitation, as shown in Line 112, we will add the meaning of "P" in formula 4.

Line 193: "more than 882.36". If you provide such a specific value indicating the amount of transpired water, I expect to read "it was 882.36 mm" or "it was more than 880 mm".

**Authors response:**

Thank you for your detailed comments. Some meteorological data was not collected because of instrument maintenance (Lines 114-115). The $ET_0$ in 2020 was calculated based on the accessed data, which was 882.36 mm. We will rewrite the sentence as "more than 880 mm in 2020" in the revised manuscript.

Line 205-206: "Both plantations increased SWS from shallow to deep soil layers over time in 2020". Looking at figure 3, it seems that there is a decrease in SWS for R. pseudoacacia between 0 and 100 cm. Please verify the data and revise the sentence.

**Authors response:**

Thanks for your detailed comments. "Both plantations increased SWS from shallow to deep soil layers over time in 2020" means the two plantations increased SWS in the shallow soil layers (e.g., 0–100 cm for *R.* plantation and 0–40 cm for *P.* plantation) during July-August, and the SWS of deep soil layers were increased during August-October (e.g., 100–200 cm for *R.* plantation and 20–200 cm for *P.* plantation), as shown in Figure 3. We will rewrite this sentence more specifically for clarify in the revised manuscript.

Figure 3: What does the range in the x-axis indicate? It varies in each plot and is not in agreement with the overall amount of SWS reported in each block of the figure. Without an explanation, it's confusing. Please, add more details in the Figure caption.

**Authors response:**

Thanks for your detailed comments. The x-axis means the ΔSWS over a period, such as the SWS in 0–20 cm soil layer in *R.* plantation have increased 1.03 mm during 2019/07/04–2019/08/10. We will double-check the Figure 3 and replot the Figure 3 by the bar width instead of the area, and we will add more detailed description in the Figure's caption.

Figure 4: Is the LMWL the same for the two sites? Were the precipitation samples collected only from one of the two sites? What is the distance between the two sites to assume that there is no difference in the isotope composition of the precipitation? Have you tested this claim?

**Authors response:**

Thanks for your detailed comments. The precipitation samples were collected by a homemade collector, which was set in an open place near the plantation. Totally, two rainfall collectors were set to collect precipitation samples. Furthermore, the straight-line distance between the two plantations is about 700 m, and the two plantations had the similar altitude and slope aspect (Table 1). After isotope analysis, there was no difference in the isotope composition of the precipitation. We will rewrite more detailed information of precipitation sampling in Lines 121-123 in the revised manuscript.

Line 227: "For total soil water samples". What do you mean by total soil water samples? Is this the average isotope composition for the whole soil core? Can you also provide the summary of the soil water isotope composition for samples collected in 2019?

**Authors response:**

Thanks for your comments. "total" indicates the mean isotope compositions for the whole soil samples (0–200 cm). The isotope composition of water samples was provided in the Table S1 for 2019 and Table S2 for 2020, and we will add the summary description of the soil water samples collected in 2019 as the same as that in 2020 in the revised manuscript.

Line 237-240: Have you calculated the d-excess? It could provide additional information regarding the more enriched isotope composition of the soil water in the upper layers.

**Authors response:**

Thanks for your constructive comments. The d-excess of soil water samples will be calculated and presented in the revised manuscript.

Line 242-250: I assume that the results reported in this first section are related to the direct inference method. To ease the reader's comprehension of the text, I suggest specifying it at the beginning of the sentence (as in line 251).

**Authors response:**

Thanks for your warm and detailed comments. The results of direct inference approach are indeed presented in Lines 242-250, we will rewrite the sentence as "The results of the direct inference approach showed that *R. pseudoacacia* mainly absorbed water from soil layers of 40–60 and 60–80 cm in July and August in 2019…" in the revised manuscript.

Figure 6: Why is the contribution rate of soil water to R. pseudoacacia (R.) and P. tabulaeformis (P.), calculated on the MixSIAR model, based on the values of $\delta 18O$ only? Have you tested the model also for the $\delta 2H$ results?

**Authors response:**

Thanks for your comments. It has been proved that $^{18}O$ is more stable than $^2H$, and $^2H$ is easier to be depleted during water extraction. The model results based on the $\delta^{18}O$ would be more convincing in our opinion. The detailed information can be found in: https://doi.org/10.1073/pnas.2014422117.

Line 255-256: "while P. tabulaeformis still mainly absorbed water from the 0–40 cm soil layer". But the coefficients reported in Fig. 6 for the different soil layers are very similar. And later on (line 302), you describe the water absorption for R. Pseudoacacia has evenly distributed among the different layers. But coefficients are comparable (209 July R and 2019 August P).

**Authors response:**

Thanks for your detailed comments. The contribution rate was 0.271, 0.245, 0.264 and 0.22 of 0–20, 20–40, 40–100, and 100–200 cm soil layers for *P.* plantation in August in 2019. Although the contribution rate was similar between different soil layers, the soil depth of the four soil layers were different. For example, the 0–20 cm soil layers contributed 0.271 with 20 cm soil, while the 100–200 cm soil layers contributed 0.22 with 100 cm soil. From the perspective of the overall combination of different soil layers, "*P. tabulaeformis* mainly absorbed water from the 0–40 cm soil layer". However, from the perspective of per soil layers, *R. pseudoacacia* evenly absorbed soil water from the four soil layers in July in 2019. We will double-check the description of the results, and modified the statement for consistent in the revised manuscript.

Line 278-280: The meaning of this part is not clear. Can you improve it?

**Authors response:**

Thanks for your constructive comments. We are sorry for the inaccurate and incomplete description. We will add detailed references and description in Lines 278-280 in the revised manuscript.

Line 286: "while the δ18O values in the 0–40 cm soil layer became smaller". I suggest using lower/higher or depleted/enriched.

**Authors response:**

Thanks for your detailed comments. We will replace "smaller" with "lower" in the revised manuscript.

Line 288-289: According to the isotope data, can you quantify how much precipitation water infiltrated in the soil? Is it comparable to the overall amount of precipitation for the indicated period in the area?

**Authors response:**

Thanks for your detailed comments. The amount of precipitation infiltration can be calculated by the △SWC before and after rainfall events and the infiltration depth (tracked with $\delta^{18}O$). However, our research mainly studied the changes of △SWS over a month and seasonal variation, and did not study the specific rainfall events. In our opinion, the infiltration amount would be less than overall amount of precipitation for a specific period, mainly due to soil evaporation and plants water consumption.

Line 294: Is there any kind of foliar cover on the top of the soil that could limit evaporation or is it bare soil in both the sites?

**Authors response:**

Thanks for your detailed comments. Both plantation has litter cover on the top of the soil, with litter mass of 700 and 1200 $g/m^2$ for *R.* and *P.* plantation, respectively. We will add this information in the "M&M".

Line 311: "However, the root length". The use of "however" seems inappropriate here.

**Authors response:**

Thanks for your detailed comments. We will delete "However" in the revised manuscript.

Line 337: "increasement" → Please substitute with increase or increment.

**Authors response:**

Thanks for your warm and detailed comments. We will replace "increasement" with "increment" in the revised manuscript.

Line 366-368: This part is not clear, please revise.

**Authors response:**

Thanks for your comments. The description in Lines 362-368 pointed out the pros and cons of widespread planting of both the two tree species. We will rewrite this section for clarify in the revised manuscript.

Line 371: "which would also good for the long-term and sustainable development of forest ecosystems". The verb is missing.

**Authors response:**

Thanks for your detailed comments. This sentence will be rewrite as "which would also be good for the long-term and sustainable development of forest ecosystems". We will double-check our statement throughout our paper, thanks again for your warm and constructive comments.